# Development of Focusing Plasma Mirrors for Ultraintense Laser-Driven Particle and Radiation Sources

**Robbie Wilson** [1] , **Martin King** [1] , **Ross J. Gray** [1], **David C. Carroll** [2], **Rachel J. Dance** [1], **Nicholas M. H. Butler** [1], **Chris Armstrong** [1,2], **Steve J. Hawkes** [2,1], **Robert J. Clarke** [2], **David J. Robertson** [3], **Cyril Bourgenot** [3], **David Neely** [2,1] and **Paul McKenna** [1,*]

[1]  SUPA Department of Physics, University of Strathclyde, Glasgow G4 0NG, U.K.;
    robbie.wilson@strath.ac.uk (R.W.); m.king@strath.ac.uk (M.K.); ross.gray@strath.ac.uk (R.J.G.);
    rachel.dance@strath.ac.uk (R.J.D.); nicholas.butler.2014@uni.strath.ac.uk (N.M.H.B.);
    chris.armstrong@strath.ac.uk (C.A.)
[2]  Central Laser Facility, STFC Rutherford Appleton Laboratory, Oxfordshire OX11 0QX, U.K.;
    david.carroll@stfc.ac.uk (D.C.C.); steve.hawkes@stfc.ac.uk (S.J.H.); rob.clarke@stfc.ac.uk (R.J.C.);
    david.neely@stfc.ac.uk (D.N.)
[3]  Department of Physics, Durham University, South Road, Durham DH1 3LE, U.K.;
    david.robertson@durham.ac.uk (D.J.R.); cyril.bourgenot@durham.ac.uk (C.B.)
**\***  Correspondence: paul.mckenna@strath.ac.uk; Tel.: +44-(0)141-548-5712

**Abstract:** Increasing the peak intensity to which high power laser pulses are focused can open up new regimes of laser-plasma interactions, resulting in the acceleration of ions to higher energies and more efficient generation of energetic photons. Low f-number focusing plasma mirrors, which re-image and demagnify the laser focus, provide an attractive approach to producing higher intensities, without requiring significant changes to the laser system. They are small, enhance the pulse intensity contrast and eliminate the requirement to expose expensive optics directly to target debris. We report on progress made in a programme of work to design, manufacture and optimise ellipsoidal focusing plasma mirrors. Different approaches to manufacturing these innovative optics are described, and the results of characterisation tests are presented. The procedure developed to align the optics is outlined, together with initial results from their use with a petawatt-level laser.

**Keywords:** plasma optics; plasma mirrors; laser-driven proton acceleration; radiation sources

## 1. Introduction

Since the first practical demonstration of the laser, there has been a constant push in the development of new technologies to increase the achievable peak intensity, resulting in an approximate increase of between two and three orders of magnitude per decade. Intensities exceeding $10^{20}$ Wcm$^{-2}$ are now routinely achieved at a number of high power laser facilities. Each significant increase in intensity has opened up new avenues of research in laser-matter interactions, including, for example, laser-driven particle acceleration [1,2], high energy radiation sources [3,4] and the generation of states of warm dense matter [5,6].

To exploit the transformative potential of laser-driven sources and to open up experimental investigation of ultraintense ($>10^{22}$ Wcm$^{-2}$) laser-plasma phenomena, such as high field physics [7,8], characteristic parameters of the drive laser must be improved beyond the present state-of-the-art; specifically, the peak intensity and pulse intensity contrast. An approach commonly employed involves increasing the laser pulse power, either through increasing the energy or decreasing

the duration [9]. This is the route being taken by multi-petawatt (PW) laser facilities, which are due to come on-line in the next few years (e.g., APOLLON (Palaiseau, France) [10]; the Extreme Light Infrastructure (ELI) (Dolní Břežany, Czech Republic and Magurele, Romania) [11]) and which aim to deliver peak intensities in the range $10^{22}$–$10^{23}$ Wcm$^{-2}$. This approach involves the use of large beam diameters, so that the energy density on the laser chain optics is kept below the damage threshold, which inevitably leads to relatively expensive and large laser systems.

As an alternative approach, intensity increases can be accomplished through the reduction of the focal spot size, by implementing a low F-number (F/#) focusing optic. This can be somewhat more effective than increasing the peak power as intensity scales inversely with the square of the focal spot size. Such optics are, however, typically costly and problematic to manufacture. Additionally, low F/# optics are susceptible to damage from solid target debris due to their short focal length (and therefore close proximity to the laser-target interaction). Clearly, this approach comes with an element of risk, with measures having to be taken to protect the optic. There is thus a need for new types of optical components that can circumvent this issue.

One promising approach is to use plasma-based optics to focus the laser light. The development of single-use, disposable plasma optics enables many of the short-comings discussed above to be avoided. Crucially, plasma mirrors operate at a much higher energy density and are therefore more than an order of magnitude smaller in size than conventional solid state optical components. As a result, they can be manufactured at much lower cost. Planar plasma mirrors (PPMs) [12,13] are now routinely employed at numerous high power laser facilities as a tool for suppressing laser pulse amplified spontaneous emission (ASE) and possible pre-pulses inherently present in intense pulses produced through the chirped pulse amplification (CPA) technique [14]. PPMs operate in the following way: plasma is created on the surface of a solid (typically of optical quality glass/plastic substrate), which is otherwise transparent to the laser light. Reflection of the laser pulse occurs near the plasma critical density (the electron density at which the plasma electron oscillation frequency is equal to the laser frequency). The laser intensity on the optic surface is selected such that the intensity of ASE/pre-pulses preceding the main pulse is lower than the substrate ionisation intensity threshold and are therefore transmitted through the substrate. After plasma formation at a threshold intensity, on the rising edge of the laser pulse, the remainder of the pulse is reflected, resulting in a pulse with a higher temporal intensity contrast (ratio of the peak intensity to the ASE pedestal intensity). PPMs have been employed to enable experimentation with ultra-thin (nanometre scale) target foils [15,16] and have been the subject of several dedicated characterisation studies [17–21].

The concept of a focusing plasma mirror (FPM) is similar, except that the reflecting surface of the optic substrate is curved. Through appropriate choice of the surface geometry, an incident laser pulse can be made to focus with a smaller F/#. Employing an ellipsoidal geometry with two foci enables demagnification of a focal spot to be achieved. The use of such an FPM was first demonstrated in a proof-of-principle study on a terawatt (TW)-level laser, reported in Kon et al. [22] and Nakatsutsumi et al. [23], in which F/0.4 optics were trialled. This resulted in a ×5 reduction in focal spot size compared to the spot formed by a conventional F/2.7 off-axis parabolic (OAP) mirror. The corresponding intensity enhancement was indirectly diagnosed via the measurement of the maximum energy of protons accelerated from a thin target foil positioned at the demagnified focal spot. In a prior study, building on this concept, we have demonstrated the use of FPMs with F/1 focusing [24].

Here, we report on the progress made on our programme of work to design, manufacture and test FPMs for use on a petawatt-scale laser: the Vulcan Petawatt facility at the Rutherford Appleton Laboratory (Didcot, UK). Design considerations are presented, and different approaches to their manufacture are described. The results of reflectivity and focal spot characterisation tests are presented, as are first results for the use of the FPMs for laser-driven proton acceleration. The overall aim of this work is to help bring plasma-based optical technology closer to maturity and, specifically, to help develop the concept of FPMs from demonstration towards routine use in laser-plasma research.

## 2. Optic Design

An ellipsoidal geometry was selected for the FPM optic shape (as used in the previous work [22,23]) and is depicted in Figure 1a. The ellipse possesses two foci positions, $f_1$ and $f_2$, located on the major axis, of equal distance from the shape centre, enabling point-to-point (i.e., focus-to-focus) re-imaging. Depending on the degree of elliptical eccentricity, $e$, a reduction (or enlargement) in the image size at one focus can be obtained when an object is located at the other. The magnification, $m$, is equal to the ratio of lengths $\beta/\alpha$; where $\alpha$ and $\beta$ are the distances from the mirror surface to $f_1$ and $f_2$, respectively. As this ratio changes as a function of the beam incident angle, $\theta_{in}$ (with respect to the major axis), it can be expressed as [25]:

$$m = \frac{(1 + e^2) - 2e\cos(\theta_{In})}{(1 - e^2)} \tag{1}$$

where $e = \sqrt{(1 - b^2/a^2)}$; with $a$ and $b$ being the semi-major and semi-minor axis lengths, respectively. Practically, this geometry enables a conventional off-axis parabola (OAP) to be aligned such that its focal spot coincides with position $f_1$. As the light diverges beyond $f_1$, it is reflected by the plasma it generates on the optic surface and is then brought to focus at position $f_2$. The OAP focal spot, at $f_1$, is demagnified at $f_2$ by a factor that depends on the selected FPM geometry and $\theta_{in}$.

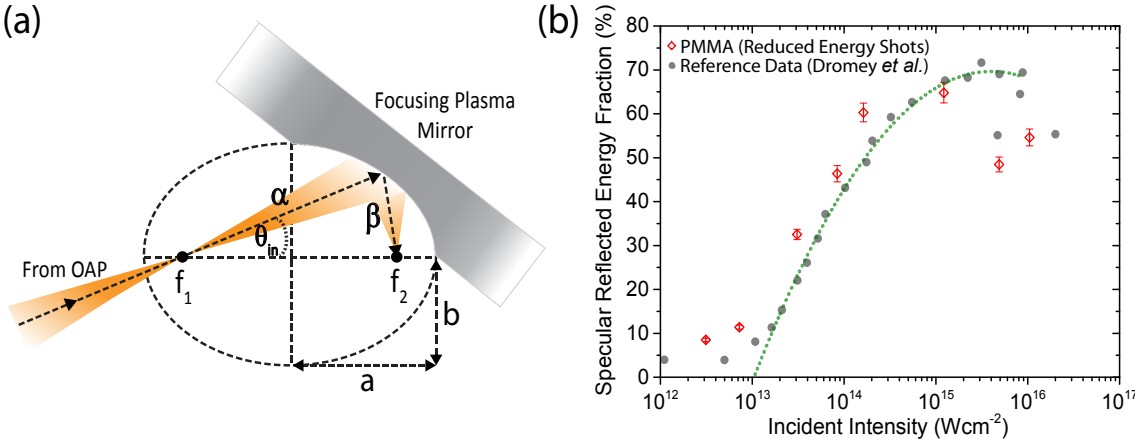

**Figure 1.** (**a**) Schematic illustrating the operation of an ellipsoidal focusing plasma mirror (FPM), where the incoming laser is focused by an off-axis parabolic (OAP) to position $f_1$ and the FPM re-focuses to position $f_2$, with magnification given by $\beta/\alpha$. (**b**) Percentage of laser light specularly reflected from a planar plasma mirror (PPM) as a function of the incident laser intensity. Red points correspond to measurements from reduced energy Vulcan-PW pulses on a flat PPM made from polymethyl methacrylate (PMMA), and grey points correspond to reference data, reported in Dromey et al. [18] (measured employing a 500 fs pulse and a fused silica PPM, with a 6° incidence angle). A quadratic fit is made to these data between $10^{13}$–$10^{16}$ Wcm$^{-2}$, as illustrated by the dotted green curve.

The exact design of an FPM is dependent on key parameters of the laser system it is intended for employment on; chiefly, the laser pulse peak power and the F/# of the OAP used to bring the pulse to focus at $f_1$. The FPM reported here has been designed for use on the Vulcan-PW laser, a system that delivers pulses of 1053 nm light, with energy ∼500 J pre-compressor (typically ∼300 J on-target including losses in the compressor) and duration ∼750 fs (full width at half maximum (FWHM)). This system employs a conventional F/3.1 OAP to focus the laser light to a typical focal spot diameter of ∼4 μm (FWHM), resulting in a calculated peak intensity of ∼6 × $10^{20}$ Wcm$^{-2}$ (assuming 30% of the energy is contained within the focal spot FWHM area).

The first step in the FPM design is the selection of the desired demagnification and, thus, the intensity enhancement factor. Although the smallest focal spot possible is desired, extremely small F/# optics are highly sensitive to alignment. A compromise value is selected, which produces significant intensity enhancement whilst enabling ease of use and robustness to a non-optimum alignment. A demagnification of $\times 3$ ($m = 1/3$) was selected based on the F/3.1 OAP, such that the FPM will yield a focal spot size of $\sim$1.3 µm (FWHM). The specific dimensions of the FPM depend on maximising the reflectivity of the plasma optic and therefore the incident laser intensity on the optic's surface (the plasma reflectivity dependence on laser intensity is reported in Dromey et al. [18]). A high specular reflectivity ($\sim$70%) is established at an incident intensity of $\sim$$10^{15}$ Wcm$^{-2}$, as shown in Figure 1b. The laser intensity at the optic surface is determined by the beam area and thus the distance the beam expands from $f_1$, i.e., $\alpha$ in Figure 1a. Through Gaussian beam expansion, this distance is determined such that an intensity of $10^{15}$ Wcm$^{-2}$ is achieved on the FPM surface, resulting in a high specular reflectivity. For alignment purposes, the incidence angle $\theta_{In}$ is selected to be 19.4° for the desired demagnification, along with a minimum distance of position $f_2$ from the ellipsoidal surface for practical target placement. Using these design characteristics, the remaining parameters required to define the optic geometry are obtained using simple trigonometry.

## 3. Optic Manufacture

The two techniques used to manufacture the FPMs will be discussed in this section: (i) injection moulding; and (ii) diamond machining. A method of manufacture is required that not only produces high quality optics (i.e., high accuracy to design and an optical quality surface), but ideally also enable volume production. The latter would enable large numbers of laser shots to be obtained and may provide a route towards use on relatively high repetition rate lasers. The quality of the focal spot formed by the optics is determined not only by the accuracy of alignment of the OAP input spot to position $f_1$, but also the quality of the optic surface, both in terms of curvature and surface roughness. Thus, selection of a suitable manufacturing technique is crucial.

### 3.1. Injection Moulding

Injection moulding techniques should in principle enable a large number of optics to be produced moderately quickly, in a reasonably cost-effective manner. The procedure we adopted for this was as follows. Firstly, a metallic mould of the FPM design (Figure 2a) was created using ultra-precise diamond machining (using a Moore Nanotech 250 machine (Swanzey, NH, USA)). Next, a transparent plastic (Acrypet VH-001 PMMA) in molten form was injected into the mould. A number of tests was conducted altering injection parameters (such as fill speed, plastic melt temperature, mould temperature and packing pressure) to achieve optimised filling and the best reproducibility. The optics produced (Figure 2b) have a near-constant thickness of 4 mm, ensuring the optics is rigid, making it less prone to distortions due to stresses induced by mounting.

Metrology was performed on the moulded optics' surface to determine mould replication accuracy and reproducibility between optics. This was achieved by employing a 3D profiling system, consisting of a chromatic confocal sensor (with a precision of 0.01 µm), to precisely measure the surface profile across the optic (Precitec CHRocodile system (Neu-Isenburg, Germany). In order to measure the complete form of the moulded surface, it was necessary to first gently roughen the surface using an abrasive medium (Abralon 2000 grit pads). This procedure necessarily removes a small amount of material (of order 100 nm), which has the potential to perturb measurements, especially if performed non-uniformly. As such, characterisation results must be interpreted with this in mind. Measurement of the optic form error was achieved by scanning each optic to produce an *xyz* data grid of the optic profile. For each data point, a least-squares fit is performed using the 'ideal' surface data as a reference. This study showed a significant non-uniform deviation, with a root-mean-square (RMS) variation of up to 5.4 µm (+14/−20 µm peak-to-valley). This result is expected from the injection moulding process due to significant shrinkage and warping when compared to the mould surfaces (i.e., the desired FPM

design). To correct this issue, a compensated mould design is required to account for plastic shrinkage and thus accurate reproduction of the FPM design. However, the simulation tools for predicting these deviations are inadequate at the tolerances required for the FPM design (most injection moulding is concerned with much less stringent tolerances).

Additionally, this 3D profiling characterisation is capable of determining the reproducibility between multiple optics. This should be considered the most important factor in determining the viability of the moulding approach. A number of test parts were measured to determine form reproducibility. Each measurement was put through a fitting procedure to register the parts consistently to the same coordinate basis. Comparison was then performed by selecting one of the xyz data points as the base case and comparing the difference between this base and the other points within the batch of tested optics. The RMS of the difference between the base point and each of the other measured points was calculated as a simple figure-of-merit. A mean RMS variation over the scanned region of 0.72 µm was measured. As these optics were to be used on the Vulcan-PW laser (operating at $\lambda_L = 1.053$ µm), this RMS variation in reflection is therefore on the order of the $\lambda_L$. Achieving optical performance close to the diffraction limit requires wave-front variations far less than the laser wavelength, typically of the order $\lambda_L/10$. As such, unless a significant improvement in the reproducibility of the moulded optics can be achieved, this manufacturing approach is not viable at present capabilities, irrespective of mould shape compensation to resolve the measured shape errors.

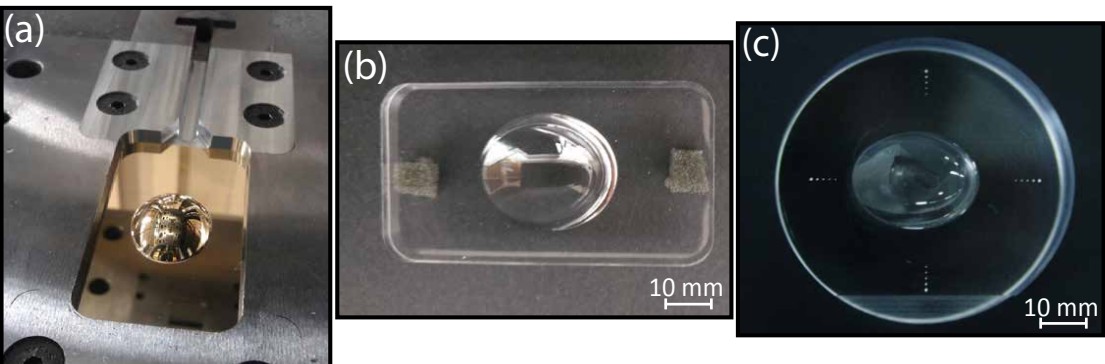

**Figure 2.** Photographs showing: (**a**) the ultra-precise machined mould used in the injection moulding process (mounted in the injection moulding tool); (**b**) an example injection moulded FPM; (**c**) an example FPM produced by diamond machining.

## 3.2. Diamond Machining

Owing to the limitations of the injection moulding manufacturing process, ultra-precise diamond machining was selected to directly manufacture each FPM optic. Diamond turning is a technique capable of producing sub-micron accuracy, even on steep free-form surfaces, as in the case of the FPM design. Figure 2c shows a picture of an optic produced using this approach. The ellipsoidal shape was machined into the middle of a two-inch diameter reference flat substrate (machined flat), to ensure accurate design reproducibility. As in the injection moulding case, transparent PMMA plastic was selected as the FPM material. The optic diameter was selected for mounting in a conventional optic mount. A thickness of 20 mm was chosen to provide sufficient rigidity, minimising distortions induced via mounting.

The path the machining tool takes while cutting the optic design is of key importance in minimising tool position errors. For example, the use of a circular path when forming an elliptical aperture is not optimal as cutting will be interrupted (i.e., the ellipsoid shape is extended beyond the aperture to allow a smooth tool path trajectory, leading to an intermittent contact between tool and surface). Furthermore, a circular tool path leads to a larger axial tool displacement and the potential for increased shape error. As such, the optic manufacturers used a self-developed software program to generate an optimised tool path, which minimises tool movement by following lines of equal height. The result is a tool path starting with a circular spiral trajectory from the edge of the part when machining the flat area and gradually developing into an elliptical spiral path when approaching the optical ellipsoid design. The tool never leaves the surface and thereby minimises production time and the machine's axis motion amplitude. The ellipsoid shape has been designed to have a zero gradient at the centre to avoid possible cutting artefacts induced by residual tool offset and tool height, as reported in Bourgenot et al. [26]. The following section presents the optical testing of the FPMs manufactured employing this process.

## 4. Optic Testing

This section presents the testing conducted to characterise the FPMs manufactured through the diamond machined approach. Specifically, the optic reflectivity, in both solid state and plasma operational mode, the demagnification and the focal spot quality achieved are characterised.

### 4.1. Reflectivity Testing

PPMs typically include an anti-reflection (AR) coating, of reflectivity $\sim$0.3% [27], which acts to increase the temporal intensity contrast improvement ability. The achievable contrast enhancement factor is equal to the ratio of the plasma reflectivity to the cold reflectivity (i.e., non-plasma state, behaving like a conventional partially-reflecting solid-state optic) [28]. The FPMs reported here did not include an AR coating at the time of testing, and thus, characterising their cold reflectivity is critical to gauge how they perform as plasma mirrors. This was achieved using a spectrophotometer to measure the reflectivity of 1053 nm p-polarised light from the optic substrate over a range of incident angles, from $25°$ to $45°$, to encompass the full range of illumination angles of the diverging F/3.1 focusing beam of the Vulcan-PW laser. The average cold reflectivity was measured to be $(4.2 \pm 0.3)$%.

To characterise the plasma reflectivity, the Vulcan-PW laser was used to investigate the reflectivity of a transparent PMMA PPM (the same material as the FPM), as a function of incident laser intensity; thus checking that the selected switch-on intensity value gives high reflectivity. The PPM was irradiated with p-polarized pulses, relative to the PM surface, at a $35°$ incident angle (i.e., the same as the operational incident angle of the FPM design). The intensity was controlled by the variation of the distance between the optic surface and the laser focus, at which a peak intensity of $\sim10^{16}$ Wcm$^{-2}$ is achieved using $\sim$0.25 J pulses. These reduced energy pulses enabled for a relatively quick succession between laser shots. The energy of the incident and reflected light was measured using a Gentec pyroelectric energy meter (for absolute calorimetry). The specular reflectivity of the PPM as a function of incident laser intensity is shown in Figure 1b, where it is compared to the results reported in Dromey et al. [18]. A peak specular reflectivity of $(65 \pm 2)$% was measured at an intensity of $(1.2 \pm 0.3) \times 10^{15}$ Wcm$^{-2}$. The overall trend of the reflectivity as a function of incident intensity is in excellent agreement with the data employed in the FPM design [18]. For the measured peak plasma specular reflectivity (65%) and for the cold reflectivity measured (4.2%), the optics is expected to increase the intensity contrast by a factor of 15.5 (approximately an order of magnitude less than an AR-coated PPM).

### 4.2. Focal Spot Characterisation

An experimental set-up employing a low-power continuous wave (CW) laser was developed, shown in Figure 3a, to characterise the diamond-machined FPMs in terms of focal spot reduction and

quality and consistency between optics. This was not only employed for FPM characterisation, but additionally for the pre-alignment of optics prior to use on the Vulcan-PW laser (discussed further in Section 4.3). In this low-power illumination mode, the optic is not ionised and therefore acts simply as a conventional partially-reflecting solid-state optic.

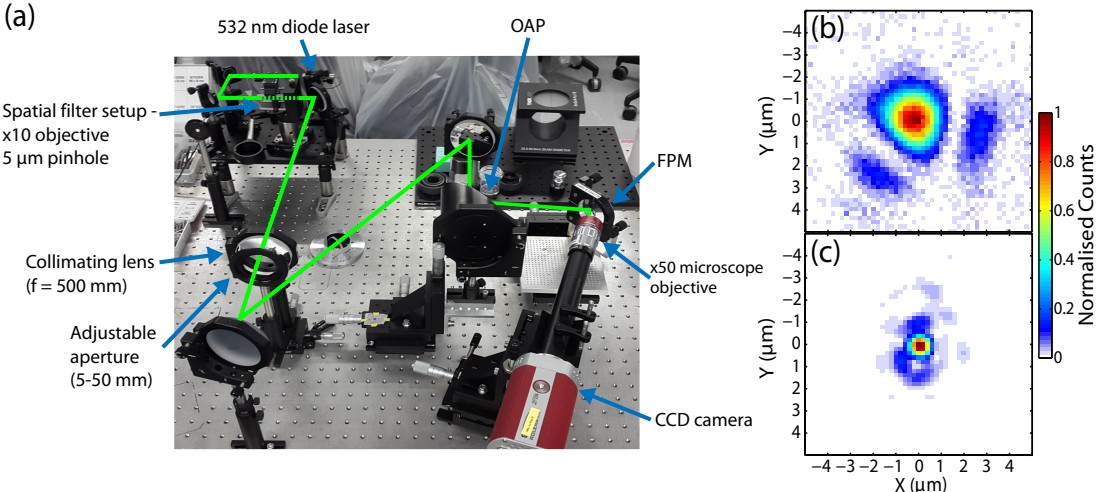

**Figure 3.** (**a**) Photograph of the experimental set-up developed to characterise the FPMs in non-plasma operation, where the main components are labelled and the laser path is shown in green. Measured focal spot spatial-intensity distributions using the characterisation set-up at: (**b**) $f_1$ (input focus); and (**c**) $f_2$ (output focus).

An OAP (f = 145 mm) and a 48 mm diameter collimated input laser beam was used to emulate the Vulcan-PW input focusing beam (F/3.1). A 532 nm laser diode was used as the light source, with the beam first propagating through a spatial filter to improve its spatial-intensity profile. This wavelength of light was selected as it is shorter than the normal FPM operational wavelength (1053 nm) and thus aids in determining if there are any undesired irregularities in the optic operation. These are displayed more prominently at the lower wavelength as it has a smaller diffraction limited spot size for a given F/#. To characterise the focal spots formed by both the FPM and OAP, an infinity-corrected microscope objective ($\times$50 Mitutoyo Plan Apo NIR B) was used to image the spot to a 16-bit charge-coupled device (CCD). The field of view was 129 μm $\times$ 97 μm. This objective possesses a 25 mm working distance (separation between initial lens and image plane), significantly larger than similar magnification objectives. This is necessary to allow for the imaging of the spot formed at position $f_2$ without the objective coming into contact with the FPM, due to the focus position being close to the FPM surface. For alignment and imaging purposes, the FPM, OAP and imaging components were mounted on micrometer controlled *xyz*-translation stages. Importantly, the *z*-axis of the FPM motion was set along the direction of the OAP input beam axis, which results in the optic alignment complexity being considerably reduced. As such, the FPM was mounted at an angle relative to the input beam to ensure the correct input angle, $\theta_{In}$, was satisfied. The FPM itself was mounted using a two-inch (front mounted) optic mount to give additional motion control (tip and tilt) and the ability to rotate the optics within the mount. Once the OAP focal spot was optimised, the FPM set-up was added and aligned by translating the *xyz*-position, whilst monitoring the output focal spot (i.e., at $f_2$). The end goal of this process was an optimum alignment, i.e., when the OAP focus spatially coincides with the first FPM focus, $f_1$.

Optic characterisation was conducted by analysing the output focal spot formed by the FPM under optimum alignment. An example result is displayed in Figure 3b,c. The optic sensitivity to non-optimum alignment is reported in [24]. The typical output focal spot formed by the FPM (at $f_2$), displayed in Figure 3c, is 0.76 μm (FWHM), with 28.3% energy encircled within the FWHM diameter. Based on the input focal spot (Figure 3b), of 1.91 μm (FWHM) with 35.1% FWHM encircled energy, a focal spot demagnification of ×2.5 was achieved. The reduction in encircled energy may be attributed to small misalignments. It may also be because the calculation of magnification is under paraxial approximation (in Equation (1)), which is no longer valid when the focal spot diameter is of the order of the laser wavelength. We also note that some of the light may be diffracted out of the focal spot if the diamond-turned optics have periodic tool marks. A relatively large field of view was sampled by the microscope objective, but a fuller investigation of this potential effect could be made by measuring the energy transmitted through a pinhole.

Using these measurements, the expected enhancement factor in the laser intensity, $I_{Enh}$, under plasma operation can be calculated. This parameter depends on the input ($\phi_{in}$) and output ($\phi_{out}$) spot sizes, encircled energies ($E_{in}$ and $E_{out}$, respectively) and the optic plasma reflectivity ($\Gamma_p$), as:

$$I_{Enh} = \left( \frac{\phi_{in}}{\phi_{out}} \right)^2 \cdot \Gamma_p \cdot \left( \frac{E_{out}}{E_{in}} \right) \tag{2}$$

For the spot characterisation results and a plasma reflectivity of 70% (anticipated from the optic design), the calculated intensity enhancement is ×3.6. The minimum plasma reflectivity at which enhancement is still achieved is 19.7%, i.e., when $I_{Enh} = 1$, the point at which the intensity of the input focal spot equals that of the output. In terms of the energy contained within the output focal spot FWHM, enhancement would begin for values exceeding 7.9% for $E_{out}$, in the 70% reflectivity case.

During the characterisation process, it was found that ∼60% of the batch of optics manufactured exhibited the high quality output focal spot displayed in Figure 3c. Figure 4a displays the typical focal spot achieved with a sub-optimum quality FPM, which are referred to as Optics A, compared to the focal spot produced by the optics operating to specification (Figure 4b), which are referred to as Optics B. Prominent differences in the distributions of the focal spot energy are observed. There is a considerable drop in the energy encompassed within the central spot of Optics A (Figure 4a), and instead, significant energy is contained in multiple rings surrounding the focal spot. To quantify, Optics A forms a focal spot with 5.1% energy encircled within the FWHM diameter, a ×5.5 reduction compared to the Optics B case (28.3% FWHM encircled energy). Accordingly, intensity enhancement would not be achieved for Optics A. The wave-front of the light from both Optics A and B was was also measured. The peak-to-peak deviation with respect to an ideal wave-front was measured to be $2.65\lambda_L$ for Optics A and $0.01\lambda_L$ for Optics B, for the degree of astigmatism (at 0°) aberration.

The source of this variation between optics becomes clear when characterising the FPM output focal spot as a function of the input beam F/#. Variation of the F/# changes the divergence of the input beam and, hence, the area illuminated on the optic surface. This study was implemented in a controlled manner, using a variable diameter aperture in the test set-up (Figure 3a) to control the diameter of the collimated beam prior to the OAP. Figure 4c displays the results of changing the input F/# in terms of the resultant output focal spot encircled energy for each optic case. At small input F/#, the optics have significantly different values. With increasing F/#, the values become broadly similar. This trend suggests a shape error in the optic geometry, leading to the low quality focal spot, which is only seen when a relatively large area of the optics is illuminated (i.e., small F/#). To correct this issue, optics displaying this low quality nature (Optics A case) were re-machined to the desired shape, after which they exhibited focal spot quality similar to those achieved by Optics B (Figure 4b). This highlights the sensitive nature of the manufacturing process and the need for pre-characterisation of each optics before use.

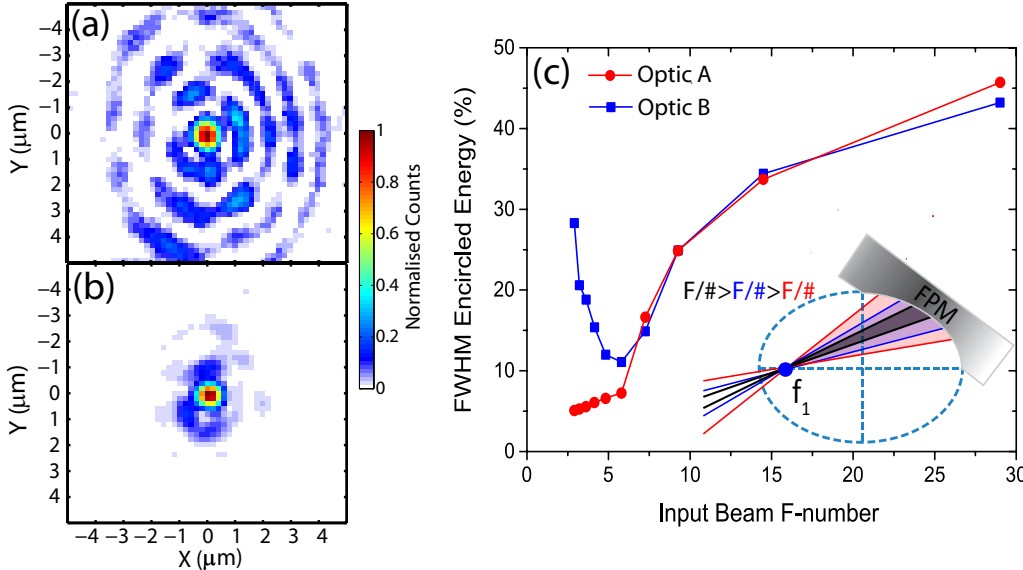

**Figure 4.** Typical measured spatial-intensity distributions of the optimised focal spot formed by: (**a**) Optics A; and (**b**) Optics B. (**c**) Plot quantifying the change in output focal spot FWHM encircled energy as a function of the input beam F-number. The red and blue curves represent results obtained for Optics A and Optics B, respectively. Insert: schematic showing the concept of varying the input beam F/# and the resultant change in the illuminated area on the FPM surface (black corresponds to the largest F/#, and red corresponds to the smallest).

### 4.3. Plasma Operation Testing

Finally, the FPM operation was tested in plasma reflection mode using the Vulcan PW-laser. The focal spot formed by the optics cannot be directly measured during a full power laser shot, and there is no easy way to filter out the high energy reflected in the rapidly-expanding beam. Instead, measurements of proton acceleration from thin foil position at $f_2$ were used to infer the intensity enhancement. Protons are produced and accelerated by a strong electrostatic field formed at the target rear surface, via the target normal sheath acceleration (TNSA) mechanism [29]. The field is generated by fast electrons produced in the laser focus and transported through the foil. The maximum proton energy is correlated with the peak laser intensity [30,31], via the temperature and density of the fast electrons [32].

Before each FPM was mounted in the Vulcan-PW target chamber, it was pre-aligned using the characterisation set-up shown in Figure 3a. This involved first optimising the FPM output focal spot, as for the case in Figure 3c, i.e., spatial overlap of the OAP focus with the ellipsoid's first focus, $f_1$. When this was achieved, it was then necessary to mark this overlapping position, as this defines the position at which the Vulcan-PW beam was required to be brought to focus. To achieve this, a glass fibre wire (7 μm diameter) was mounted on an *xyz*-translation stage and attached to the same mounting board as the FPM. This was used to define the position $f_1$; the rig was translated until the fibre obscured the laser focal spot as measured with an alignment camera. The wire set-up relative to the optics is shown in Figure 5.

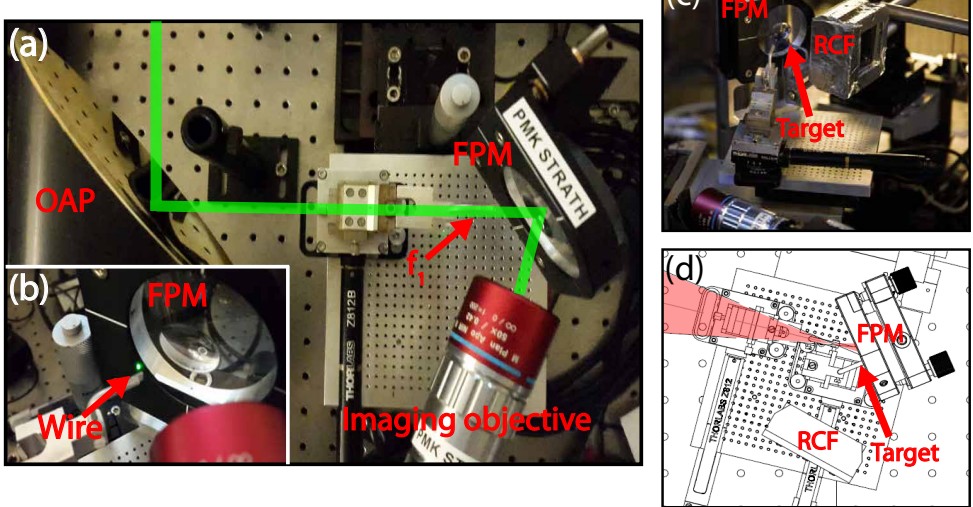

**Figure 5.** (**a**) Photograph of the set-up used to pre-align an FPM before transferring to the Vulcan-PW vacuum chamber, where a 7 μm diameter wire was used to define position $f_1$. (**b**) Photograph showing the alignment wire at position $f_1$, scattering green light from the OAP focus. (**c**) Photograph of the experimental set-up employed to quantify the maximum energy of laser-accelerated protons from a 6 μm-thick Al target positioned at $f_2$ and a stack of dosimetry radio-chromic film (RCF). (**d**) Equivalent schematic drawing.

After the pre-alignment steps were completed, the set-up board was transferred to the target chamber, where it was mounted on a high resolution *xyz*-stage (on a kinematic base) for precise FPM positioning. This provided the equivalent freedom of motion as used in the test set-up (shown in Figure 5c,d). Off-line pre-alignment to precisely define the position $f_1$ was necessary to reduce the optic alignment time and complexity within the target chamber. This enabled the total alignment time to be reduced to less than the typical time between high power shots on the Vulcan. The retro-reflection set-up on the Vulcan was employed to align the Vulcan-PW OAP focal spot to the pre-alignment wire (defining position $f_1$). This collects scattered laser light from the wire (or other targets positioned at the focus of the OAP) and focuses it down for monitoring by a CCD. The FPM was positioned such that the OAP focus was at the tip of the wire and therefore at position $f_1$. The wire was than translated out of the beam path using a goniometer stage (so as not to impede optic irradiation), and the target was moved into position at $f_2$.

Before the FPM test shots on a foil target, the optic was characterised in terms of optical demagnification and output focal spot quality in situ within the Vulcan target chamber (in a similar method as the off-line test set-up characterisation below the ionisation threshold, discussed above). Figure 6a,b shows the typical input and output focal spots achieved under optimised CW alignment with the 1053 nm Vulcan beam. Both the input and output spots are relatively larger in comparison to the 532-nm characterisation (Figure 3). The FPM produced a factor of ×2.5 reduction in the spot FWHM (from 4.0 μm input to 1.6 μm output), and the energy encircled within the area defined by the FWHM increased from 28.1% to 36.5%. These measurements are in good agreement with the characterisation using 532-nm light (Figure 3b,c). A calculated peak intensity equal to $3.4 \times 10^{21}$ Wcm$^{-2}$ could be achieved using the Vulcan-PW laser parameters, which is a factor of ×5.3 intensity enhancement over the standard F/3.1 OAP focusing.

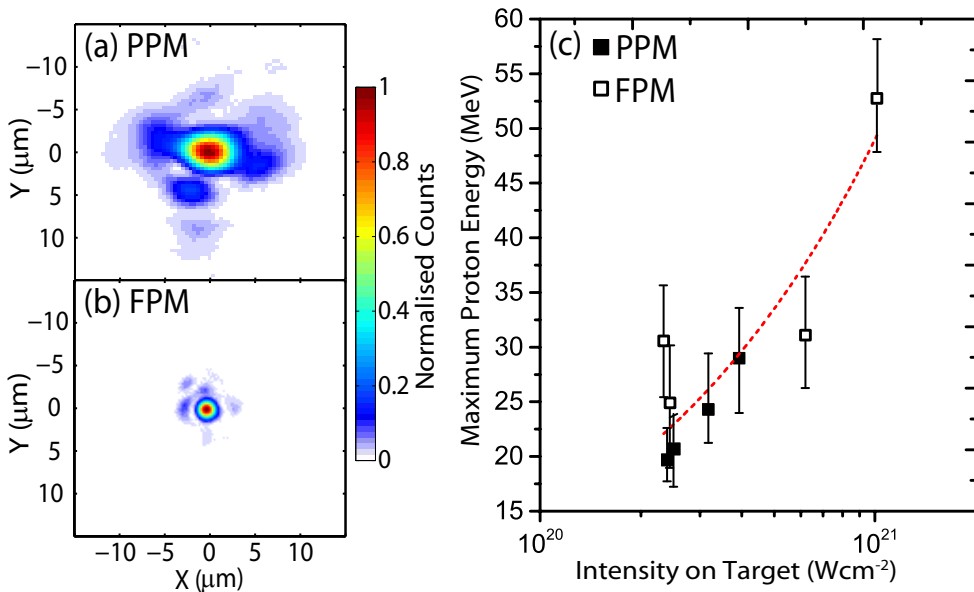

**Figure 6.** Measured laser focal spot spatial-intensity distributions using the Vulcan-PW laser, with low power (continuous wave (CW) operation) 1053 nm light, obtained for (**a**) a planar plasma mirror and (**b**) a focusing plasma mirror, at $f_2$. (**c**) Plot of the measured maximum proton energy achieved using FPMs (open squares) and PPMs (closed squares), as a function of the laser intensity ($I_L$). The red dashed line represents a simple fit of the form $E_{pmax} = a.I_L^b$, with $b = (0.6 \pm 0.1)$ (see for example [30,31].)

Finally, the FPMs were used in plasma operation mode, with full-power laser shots, to investigate laser-driven proton acceleration. Comparative proton acceleration measurements were also made using PPMs, made from the same material (PMMA). The PPM shots were necessary to acquire reference proton beam measurements with the same reflectivity and intensity contrast enhancement, for direct comparison with the FPM performance. The laser was incident at the target normal onto 6 µm-thick aluminium foils, for all shots. The beam of accelerated protons was measured using a stack of dosimetry radiochromic film (RCF) [33], positioned 50 mm behind the target foil and centred on the target normal axis. This enabled the proton beam spatial-intensity distribution to be characterised in discrete energy bands, in the range from 1.1 to 86 MeV Figure 6c shows the measured scaling of the maximum proton energy as a function of intensity, for both the PPM and FPM series of shots. Compared to the PPM, the FPM resulted in an intensity enhancement factor of ×2.6 (from $3.9 \times 10^{20}$ Wcm$^{-2}$ to $1.0 \times 10^{21}$ Wcm$^{-2}$), which increased the maximum proton energy by almost a factor of two. We note that there are also differences in the proton beam divergence and spatial-intensity distributions measured with the PPM and FPM optics, and the underlying physics giving rise to this will be addressed in a dedicated follow-up investigation.

The range of intensities produced with the FPMs resulted from a thermal lens effect in the laser amplifiers, which changed the wave front of the beam from shot-to-shot. This was measured on every shot and factored into the intensity calculation. Due to the short focal length of the FPM, the final intensity achieved is also very sensitive to small misalignments, as discussed in [24]. Due care is therefore required in both the alignment and the characterisation of the incident laser beam to ensure that intensity enhancement is achieved.

## 5. Conclusions

In conclusion, the design, manufacture and testing of a novel FPM, based on an ellipsoidal geometry, is reported. This enables tailoring of an intense laser pulse in terms of its focal spot size and temporal intensity contrast. The FPM design involved the optimisation of the incident laser intensity to maximise the plasma reflectivity. The FPM testing conducted highlights the validity of this approach.

Two manufacturing methods used to produce the FPM, namely injection moulding and diamond machining, are discussed. Both were trialled and the latter method was selected as the most viable method to produce usable optics for testing and deployment in the short term, as it is capable of producing the desired FPM design to a higher degree of accuracy and precision, albeit at relatively high cost. Further work is required to improve the injection moulding approach, such that the moulded optics are not distorted during the setting phase. If achieved, this would represent a major development, enabling the volume manufacture of low cost, disposable plasma optics.

Direct measurements of the focal spot formed by the FPM, in non-plasma operation, showed an optic demagnification of ×2.5 and a relatively high quality spot (28.3% encircled within the FWHM diameter). These measurements suggest an estimated factor of ×3.6 intensity enhancement, when considering the focal spot quality (encircled energy) and plasma reflectivity. Under optimum alignment and laser conditions (i.e., maximum pulse energy and shortest length), a calculated peak intensity of $\sim$4 $\times$ $10^{21}$ Wcm$^{-2}$ could be achieved whilst employing the FPM on the Vulcan Petawatt laser system. This characterisation study was not only necessary to gauge the performance of the optics (in terms of focal spot demagnification and quality), but importantly the reproducibility between optics (as highlighted by the varying quality of optics produced) and pre-alignment before use on the high power laser system.

Results from an example use of the FPMs in an investigation of laser-driven proton acceleration are presented. The intensity enhancement achieved by the optics was utilised to increase the maximum energy of laser-accelerated protons from thin foil targets, from 27 MeV to 53 MeV, almost a factor of two higher than F/3.1 OAP focusing and consistent with TNSA laser intensity scaling [30,31]. The highest proton energy is achieved when the optics is optimally aligned, i.e., when the OAP focal spot spatially overlaps the FPM input focus (discussed further in [24]).

This programme of work helps to bring focusing plasma optics a step closer to regular deployment in high power laser experiments, in much the same way as planar plasma mirrors are routinely deployed at present. The uptake of this technology would be greatly enhanced by additional work on developing the injection moulding or similar approaches to increase the speed and decrease the cost of manufacture. The use of these optics to extend the intensity achievable could have a significant impact on laser-plasma physics, ranging from the development of laser-driven particle and radiation sources to the exploration of new research topics, such as high field physics.

**Acknowledgments:** We acknowledge the expert support of staff at the Central Laser Facility of the Rutherford Appleton Laboratory. This research is financially supported by EPSRC (Grants EP/M018091/1, EP/R006202/1, EP/J003832/1, EP/L001357/1 and EP/K022415/1). Data associated with research published in this paper can be accessed at: http://dx.doi.org/10.15129/e778dc67-2b5e-40b5-b701-adb1caefee19.

**Author Contributions:** The FPM was designed and tested by R.W., with contributions from M.K. and P.McK. The FPM plasma operation experiment was executed by R.W., R.J.G., M.K. R.J.D. C.A, S.J.H. R.J.C. D.N. and P.McK., and was conceived and designed by R.W., R.J.G., D.C.C. and P.McK. All authors have reviewed the manuscript.

**Conflicts of Interest:** The authors declare no conflict of interest. The founding sponsors had no role in the design of the study; in the collection, analyses or interpretation of data; in the writing of the manuscript; nor in the decision to publish the results.

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
