# Peer review of "Development of Focusing Plasma Mirrors for Ultraintense Laser-Driven Particle and Radiation Sources"

_qubs, doi:10.3390/qubs2010001_

Round 1

Reviewer 1 Report

The authors report on design considerations and some experimentation of ellipsoidally shaped plasma mirrors intended to enhance peak laser intensity by further focusing a converging beam once it is near to the target. Such plasma mirrors are therefore single-use, and so the authors have investigated more than one manufacture process in an attempt to make these optics faster and cheaper to produce. There is also discuss of the careful alignment procedure necessary to use these devices, as well as some initial experiments with them form the Vulcan Petawatt laser.

The authors have outlined the necessity and benefits of focusing plasma mirrors quite well, and have done a commendable job illuminating details of the two manufacturing techniques for the mirrors. It appears this manuscript in general offers additional details extending work previously published in the cited reference [24], Wilson et al. PoP 2016. I would suggest that more detail on several aspects be added to further distinguish this manuscript from that earlier work. In particular:

1)      The surface details of the injection moulding optics are presented, and it is pointed out that the variation is too large for these to produce a high quality laser focus. It is clear from the reflected spot sizes in Fig. 3 that the diamond turned mirrors have superior surfaces, but it would be beneficial to include some of the same measurements done for the injection moulding case to drive home the comparison. Another nice figure could be a wavefront sensor measurement of the reflection from the plasma mirror, if that was performed.

2)      Relatedly, could the surface quality differences between Optic A and B (the different qualities of diamond turned mirrors) be quantified? From a manufacturing perspective it would be interesting to know what surface quality is sufficient for a given encircled energy/ focal spot size upon reflection.

3)      The authors mention that the intrinsic laser wavefront was taken into account during the experiment and their calculation of the on-target intensity was appropriately adjusted; was the possible difference in the plasma mirror wavefront also accounted for (i.e. was the wavefront measured after plasma mirror reflection)? Relatedly, were any Optic B mirrors shot in this fashion, and those ion results compared to the optics with superior surface quality?

4)      The authors mention the plasma mirror was adjusted to minimize the CW alignment beam from the Petawatt—is it possible the focusing plasma mirrors were compensating somewhat for wavefront errors coming from the amplifier thermal lensing? A quantification of this sort of effect would be of interest to facilities whose repetition rate is limited by such effects, and could use these mirrors to increase their shot cycle.

5)      Does the RCF (or perhaps other diagnostics) reveal ion differences between the planar and focusing mirror shots other than maximum energy, e.g. any noticeable difference in ion source size?

Author Response

We thank the reviewer for the very positive assessment of our manuscript and address each of the reviewer’s comments below.

1)    The surface quality of the injection moulded optics was found to be significantly inferior to that of the diamond turned mirrors and therefore a focal spot characterisation was not performed for the moulded optics. Wavefront sensor measurements of the reflected light were only made (in low power, non-plasma reflection mode) for the diamond turned optics (discussed further in response to point 2).

2)    The surface quality of Optic A (the sub-optimum quality focusing plasma optic) was not characterised in detail and unfortunately this optic was subsequently re-machined, and is therefore no longer available to perform this measurement. However, the wavefront of the light from both Optic A and B was measured (in low power, non-plasma reflection mode) and these values have been added to the manuscript (line 260 on page 8). The peak-to-peak deviation for the level of astigmatism (at 0 degrees) with respect to an ideal wavefront was measured to be 2.65Lambda_L for Optic A and 0.01Lamda_L for Optic B, where Lambda_L is the laser wavelength.

3)    The wavefront could not be measured after plasma mirror reflection in full power, plasma operation mode, because the beam rapidly expands after focus and there is no straightforward method to dump the excess energy which would otherwise damage the wavefront sensor CCD. The optics with the sub-optimal surface quality were not shot in full power mode. They were instead re-machined to higher quality (due to the limited numbers of optics and laser shots available), and therefore there are no measurements of ion acceleration from the sub-optimal optics.

4)    We do not have evidence available, from the limited number of shots that we had, to confirm or otherwise whether the focusing plasma mirrors help to compensate for wavefront errors from the amplifier thermal lensing. We agree with the reviewer that thermal lensing can be problematic on high energy laser facilities and can limit the shot rate, and that a technique to compensate for this effect would thereby certainly be of interest. This could be investigated in a future experimental campaign with more optics and laser shots available to perform a focal spot scan.

5)    There are differences in the proton beam profiles measured with the planar and focusing mirror shots. However, the underlying physics responsible for these differences is not yet fully understood. We are investigating this at present and it will hopefully form the basis of a future, dedicated paper on proton acceleration with tightly focused laser pulses. We have added a sentence to the revised manuscript (line 330 on page 11) to indicate this.

We again thank the reviewer for the positive review of our work.

Reviewer 2 Report

The paper is technically sound and important stepping stone towards achieving high intensity on current generation laser facilities. I would recommend the paper to be published after some minor additions -

Diamond turn optics tend to diffract light because of periodic tool marks. Thus the calculation of energy enclosed at focus after a diamond turn optic should include a very wide field of view. What is the field of view with the 50X objective? Ideally you would like to calculate the energy enclosed over a few mm area. Another way of doing so would be to align a pin hole (for e.g., 50 um diameter) at the focus and measure the power transmitted. The amount of scattered energy can be estimated by comparing it to power measured without pin hole. This will give a calibration factor on how much energy is scattered. I feel this is an important contribution that needs to be accounted for. I hope the authors have a diamond turned FPM to make this measurement and incorporate it in the paper. In case they do not have any optic remaining this effect should be at least acknowledged in the paper. Do the authors have any comment on how this scattering will effect when the FPM is used in a real shot?

The observed de-magnification of 2.5 is very close to the design value of 3. The discrepancy can be because of the fact that the calculation of magnification is under paraxial approximation. This calculation is not valid when the output focal spot diameter is of the order of wavelength of light - which is the case for the authors. A comparison with non-paraxial calculation is beyond the scope of this paper. But the authors can mention the possible role of paraxial approximation in the discrepancy of the measured de-magnification.

Line 222. Should "relativity" be changed to "significantly"?

Line 125. The authors should include the precision of the chromatic confocal system.

Author Response

We thank the reviewer for the very positive assessment of our manuscript and address each of the reviewer’s comments below.

1)    Unfortunately we don’t have a focusing plasma optic available to make the suggested measurement with a pinhole. However, the field of view of the x50 objective was 129 um x 97 um (now stated in the revised manuscript on line 220 on page 7), which is very wide compared to the focal spot diameter of only 0.76 um (FWHM). So the calculation of enclosed energy is already performed over a relatively large area. This field of view was set by the largest CCD chip that we have available. As suggested by the reviewer, the possible effect of light diffraction is acknowledged in the revised manuscript (line 242 on page 7). Any scattering of energy out of the focal spot ultimately reduces the intensity achieved at the target.

2)    We agree with the reviewer’s suggestion and have now mentioned the possible role of paraxial approximation in the discussion of the discrepancy in the revised manuscript (line 240 on page 7).

"Line 222. Should "relativity" be changed to "significantly"?" - We have made this change in the revised manuscript.

"Line 125. The authors should include the precision of the chromatic confocal system." - The precision of the chromatic confocal system is 0.01 um and this is now included in the revised manuscript (line 124, page 4).

We again thank the reviewer for the positive review of our work.

Round 2

Reviewer 1 Report

I thank the authors for their clarifying comments and additions.